# Impact of COVID-19 Pandemic Lockdown on Migraine Patients in Latin America

**DOI:** 10.3390/ijerph20043598

**Published:** 2023-02-17

**Authors:** Maria Teresa Reyes-Alvarez, Ernesto Bancalari, Angel Daniel Santana Vargas, Karina Velez, Ildefonso Rodríguez-Leyva, Alejandro Marfil, Silvina Miranda, Jonathan Adrián Zegarra-Valdivia

**Affiliations:** 1SANNA/Clínica Sanchez Ferrer, Trujillo 13009, Peru; 2Faculty of Medicine, Universidad Católica Santo Toribio de Mogrovejo, Chiclayo 14012, Peru; 3Clínica AngloAmericana, Lima 15073, Peru; 4Research Department, General Hospital of Mexico “Dr. Eduardo Liceaga”, Mexico City 06720, Mexico; 5Hospital Angeles Interlomas, Mexico City 52763, Mexico; 6Hospital Central, Facultad de Medicina UASLP, San Luis Potosí 78210, Mexico; 7Neurology Service, University Hospital Dr. J. E. González, Autonomous University of Nuevo León, Monterrey 64300, Mexico; 8Instituto Central de Medicina, La Plata B1902, Argentina; 9Faculty of Health Sciences, Universidad Señor de Sipán, Chiclayo 14000, Peru

**Keywords:** COVID-19, migraine, pandemic, lockdown

## Abstract

The coronavirus (COVID-19) pandemic, confinement, fear, lifestyle changes, and worldwide health care impacted almost all diseases. Reports from countries outside Latin America revealed differences in migraine patients. In this study, we describe and compare the immediate changes in migraine symptoms associated with COVID-19 quarantine in patients from Argentina, Mexico, and Peru. An online survey was conducted from May to July 2020. The survey was answered by 243 migraine patients, with questions related to sociodemographic data, quarantine conditions, changes in working conditions, physical activity, coffee intake, healthcare access, acute migraine medication use, symptoms of anxiety, depression, and fear of COVID-19. The results show that 48.6% of migraine patients experienced worsened symptoms, 15.6% improved, and 35.8% remained unchanged. Worsening migraine symptoms were associated with staying at home during the lockdown. Intake of analgesics was associated with an increase in migraine symptoms of 18 times relative to those who did not increase their intake. Migraine symptoms improved when the number of sleep hours was increased, and we observed an improvement when patients decreased analgesic intake. The uncertainty about the end of the pandemic, the news, and social media are three items that contributed to the worsening of migraine symptoms in patients in the three investigated countries. Confinement during the first pandemic wave in Latin America harmed migraine patients who stayed home during the lockdown.

## 1. Introduction

In 2020, the COVID-19 pandemic forced home confinement for the majority of the population for several weeks worldwide, changing everybody’s lifestyle. Living under curfews and prolonged home lockdowns negatively impacted the worldwide population [1,2,3].

The pandemic had a tremendous impact on the health systems in Argentina, Mexico, and Peru due to the high flow of infected patients. Lockdown programs were put in place by the governments of these countries starting in mid-March 2020 to prevent or reduce transmission. In addition, working from home whenever possible was recommended as a form of social distancing to slow interpersonal transmission of the virus.

COVID-19 and the psychological pressure of socioeconomic impact due to everyday work (widespread in our countries), limited health resources, and the lack of knowledge of this new disease resulted in variable compliance with preventive protocols and induced worries about uncertainty regarding the near future and anxiety and fear of acquiring the disease. In addition, limited access to medical consultation and medicine refills, as well as the fear of being infected, became part of daily living. Fear as a negative emotion symptomized to avoid specific stimuli is associated with feeling personally at risk of infection from COVID-19. The potential for widespread public fear caused by a pandemic viral infection and mental distress could be seen at the population level. These changes affected migraine patients, a highly prevalent disease in our countries [1,2,3].

Several reports from other countries were published describing changes in subjects with migraine, providing physicians with insights with respect to how to help migraine patients [4,5,6]. Due to the fact that health systems and culture differ between Latin America and European countries, we wanted to understand changes that occurred during this period by analyzing trigger factors that contribute to the precipitation of migraine, how patients experienced sudden adaptive change as a result of confinement in the three investigated countries, and the impact of these variables on the worsening of migraine symptoms. On the other hand, we also believe that the fear of COVID-19 contributed to the worsening of migraine symptoms.

### Objective

The aim of this study was to evaluate the effect of COVID-19 pandemic lockdown-related trigger factors in Argentina, México, and Peru on migraine patients during the first wave of the pandemic. Herein, we answer questions regarding improved or worsened migraine symptoms during confinement, as well as changes in habits due to remote working at home, limited access to medical services, and the availability of acute migraine treatment using the modified questionnaire of fear of COVID to identify which variables had the greatest impact on the studied population, with a focus on differences between the three countries.

## 2. Materials and Methods

Survey Design: An invitation was sent to neurologist members of the Latin American Association of Headaches (ASOLAC) to participate in the study. Invitations were sent to all trained neurologists with medical experience treating headache patients and general neurology patients and with full knowledge of the current International Classification of Headache Disorders (ICHD-3) [7]. Those who wanted to participate were required to distribute an online questionnaire-based survey to their current migraine-diagnosed patients through email or WhatsApp. All the patients were known patients with medical records that fulfill the diagnosis of migraine according to the ICHD-3 with a main complaint of migraine distinct from other primary headaches (trigeminal autonomic, tension-type, and others) and with a current email or WhatsApp contact. We constructed a sample comprising patients from the three investigated countries, with more than 25 entries per country, including patients from one city in the northern and central regions of Mexico, northern and central cities in Peru, and one city in Argentina. Data were collected from May to July 2020.

The online survey was completed using Google forms. Adaptive questions were conditionally displayed based on responses to mandatory questions about worsening or improvement in migraine symptoms during the lockdowns, without a review step and without a timestamp, asking about (a) demographic data, including gender, age, country of residence, and civil status; (b) lockdown restrictions, asking how many weeks respondents were in lockdown due to government-mandated pandemic confinement in each country and how many times per week or month (on average) they left their home during the time of confinement if they were still in lockdown. (c) Migraine status: The perception of change in migraine regarding improvement or worsening in intensity and frequency during the lockdown period. (d) Acute migraine medication intake: were asked two questions: personal appreciation of more/less intake of acute migraine medication during the lockdown, and question asking the number of days/use of acute pain medication before and after lockdown. (e) Access to medical services, face-to-face or virtual during a lockdown or in the emergency room, and access to pain medication, or no availability of this service. (f) Occupation during lockdown regarding working as usual, or working in or out from home, an increased at-home workout during the lockdown, and a new routine at home (home-office, virtual remote education). (g) Exercise: Frequency before and after lockdown as changes in their physical activity. (h) Sleep habits: Regarding sleeping more hours or fewer hours, or equal, and quality of sleep compared to before lockdown. (i) Coffee intake before and during lockdown as a relative appreciation, and as a question regarding the number of cups of coffee before and after lockdown. (j) Symptoms of anxiety and depression were quantified by the two first questions of the General Anxiety Disorder (GAD7) and the Patient Depression Questionnaire (PHQ9), which is synthesized in the PHQ4 questionnaire [8]. (k) Fear of COVID-19 done by eight questions regarding fear and anxiety as per fear of coronavirus-19 modified scale (FCV-19S). It included a five-item Likert scale (agree, agree, neutral, disagree, strongly disagree) to detect fear of COVID-19 among the general and specific populations [9,10,11,12,13].

Unique site visitors were determined by participants’ email addresses and eliminating double entries, keeping only the first entry. The rate/participation proportion was unavailable, so the recruitment rate was unavailable. Incomplete questionnaires were not analyzed [14].

Informed consent: Participants were informed about the length of the survey (5 to 10 min), that their data would be anonymous, with no registration of personal identity and data protected by authorized access to assigned staff only, to a web-based automatic recollection data entry. Their participation was voluntary, without any incentives offered. All patients needed to give their consent before they could start the survey.

Testing: The previous testing of the technical functionality of electronic questionnaires had been done before sending it.

Ethical Considerations: The study was conducted following the Institutional Research Ethics and the declaration of Helsinki. Formal ethical approval was granted by the Local Ethical Committee, SANNA Clinic (Ref: SANNA/MIG/01/2020). The consent form documented the study’s aims, nature, and procedure. Anonymity and confidentially were strictly maintained.

Statistical analyses: Statistical analyses were performed with the Statistical Package for the Social Sciences (SPSS 21.0, SPSS Inc., Chicago, IL, USA) software. Quantitative variables were expressed as mean and standard deviation (SD), and categorical variables were expressed as counts and percentages. The association of variables across countries regarding improvement, worsening, and remaining unchanged were assessed with the chi-square test and post hoc analysis with Bonferroni correction. Multinomial logistic regression models were used to identify the factors influencing the worsening or improvement of migraines, including sociodemographics and lockdown conditions in our model. The questionnaire’s reliability was calculated with Cronbach’s alpha. Statistical significance in all tests was set at *p* ≤ 0.05.

## 3. Results

### 3.1. Findings during the Lockdown across Countries

When analyzing the findings across countries, we found associated responses regarding their symptoms, medication for acute treatment of migraine, depression, and anxiety symptoms, and changing house habits due to lockdown.

### 3.2. Participants and Demographics Characteristics across Argentina, Mexico, and Peru

Two hundred forty-three migraine patients fulfilled the online survey named *Encuesta: Cambios en migraña durante la cuarentena (Survey: Changes in Migraine during Quarantine).* Most respondents were between 30 to 50 years, female, with civil status related to cohabiting and occupation out of the home before lockdown. More than 70% of the respondents were still in lockdown when they answered the questionnaire. More patients in Peru were in lockdown and with restrictions to go out of home (Table 1).

### 3.3. Associated Factors for Improvement and Worsening of Migraine Symptoms

Overall, there were more parameters associated with Worsening than Improvement versus remaining unchanged. The multinomial logistic regression models show the regression coefficient and OR “Exp(B)” for each of the variables compared to their respective reference value as follows: demographic and type of confinement during the pandemic period (age, civil status, and months in lockdown, still in lockdown, lockdown restriction, and indoor/outdoor work before lockdown), changes caused by confinement (sleep quality, workout, daily routine, use of analgesics) and the emotional status related to anxiety/depression and fear to COVID-19. See Table 2.

The covariates related to the demographic data showed no chance of improvement or worsened migraine if patients were cohabitating or single. Age was related to worsening (OR = 2.049, 95% CI: 1.05–3.99; *p* < 0.05) in the 30 to 50-year-old group compared to the 50+ age group. There was no difference in those younger than 30 years. Covariates related to lockdowns, such as the pandemic period and following recommendations of staying at home, did not influence migraine symptoms. The pandemic changed the workplace for many people by changing the work setting to work at home, increasing the severity of migraine symptoms (OR = 3.746, 95% CI: 1.68–8.35; *p* < 0.005).

The second group of covariates included sleep quality, exercise, change in routines at home, and analgesic use as behavioral covariates. The COVID-19 Pandemic has seriously and significantly impacted sleep quality and contributed to worsening migraines and the development of sleep disorders. In the present study, the perception of poor sleep quality was current in the worsening migraine group (OR = 2.366, 95% CI: 1.38–4.03; *p* < 0.005). Moreover, analgesics are relevant in treating migraine, and multiple circumstances favor their abuse. The pandemic strongly contributed to the abuse of analgesics in the worsening group, showing the most pertinent effect overall significant covariates of the present study (OR = 14.89, 95% CI: 7.07–31.34; *p* < 0.001).

On the other hand, a few covariates were associated with improving migraines. For example, people who adapted their home routine to confinement conditions, such as drinking less coffee and modifying family activities, improved their symptoms (OR = 2.05, 95% CI: 1.08–5.78; *p* < 0.005). One relevant activity was increased exercise, which positively affected the group that improved migraine symptoms (OR = 2.6, 95% CI: 1.5–4.52; *p* < 0.005).

In the third group of covariates, regarding mood and fear of COVID-19, we found that anxiety and depression increased markedly, as showed by the PHQ-4 questionnaire scores, revealing an association with the worsening group (OR = 1.22, 95% CI: 1.06–1.41; *p* < 0.005). Regarding fear of COVID-19, we first demonstrate that the test developed by Ahorsu [13], the “Fear to COVID-19 questionnaire,” displays adequate internal consistency measured with Cronbach’s alpha (0.807) in our study population. The unidimensional structure of the questionnaire is stable and has excellent psychometrics across countries. The reliability of this modified version of the questionnaire was assessed during data acquisition [12,13]. In the logistic regression analysis, the fear of COVID-19 revealed a significant but poor effect on worsening migraine (OR = 1.058, 95% CI: 1.002–1.116; *p* < 0.05). This contribution was expected because the fear of COVID emerged with the pandemic and evolved along with suppression measures taken during the first wave.

## 4. Discussion

The current study explored the conditions and changes made by people with migraine during the first wave of the COVID-19 pandemic. The web-based questionnaire considered modifications made in the living conditions during the lockdown and the factors related to worsening or improvement at the confinement’s beginning in Latin America’s three countries: Argentina, Mexico, and Peru.

Across countries, the study sample consisted mainly of 80% women between 30 and 50 years of age, similar to other migraine study samples [4]. They cohabitate in Mexico and are almost equally single or coupled in Argentina and Peru. Before the pandemic, their daily activities were outside the home. The time-lapse of the main lockdown was more than two months.

Peru shows answers polarized to no lockdown at all, not taking into account the confinement or, on the other hand, taking it very seriously and remaining strictly indoors. In Mexico and Argentina, people did not follow recommendations regarding restrictions on going outside. Nearly 60% cope with going outdoors to do daily activities, instead of lockdown, compared to 40% in Peru.

The perception of worsening migraines was higher in participants from Peru than in Mexico or Argentina. Peruvians had more days in pain, increased intensity, and increased number of migraine-associated symptoms. The more restrictive confinement style, the higher proportion of people staying indoors, and the sudden switch to the home office might explain the significant increase in symptoms and the marked tendency to have more days with pain and intensity.

Groups were divided into those patients that did not require medical access, those who needed but had no access, and those that had access being remote (most of them) or face-to-face visits to medical services. Half of the sample of Mexicans and Argentinians did not need medical assistance during the lockdown, compared with a quarter of the Peruvians. Ten percent in Mexico and 20% in Peru needed medical attention because of worsening migraines. Still, they did not have healthcare access, and 50% percent of Argentinians and Peruvians, and 40% of Mexicans did have access to medical services.

Access to medication for acute migraine pain was similar in Mexico and Argentina, where almost 90% of patients had a stock of drugs, compared to 70% in Peru. Interestingly, 53% of Peruvians increased their intake of analgesics, followed by Argentinians (40%) and Mexicans (33%). Across countries, 44% of patients increased medication use, whereas 40% remained the same, and 16% reduced medication intake.

We also considered lifestyle modifications due to the pandemic and containment measures that might impact migraine symptoms. For example, shifting the work setting to work at home, childcare, daily habits, or space restrictions. Our study assessed lifestyle by the following variables: number of coffee cups, changes in exercise hours, and daily activities. Across countries, 40% stopped working out, 42% kept the same routine, and 17% increased their exercise. Coffee intake did not change by 67%, 20% took less, and 13% increased their intake. In general, Argentinians changed their home routine less (36%) than Mexicans (57%) or Peruvians (46%), and the adaptation to the daily routine and increase in exercise improved migraine symptoms.

### 4.1. Working from Home Increases Migrain

We found that patients doing remote work worsened their symptoms compared to those working at their offices, as shown in Table 2 (OR 3.65, *p* < 0.001). In addition, the number of days in lockdown or government restrictions impacted worsening migraine.

Disrupted basic education service or virtual schooling, lack of childcare, increased domestic violence, and reduced income has been reported to impact physical and psychological health during the first lockdown [15].

This finding differs from other European Studies, where migraines improved during lockdown [4], considering the hypothesis that scaling down demanding social lives could give freedom to organize their time [5].

It is already known that migraines can coexist with other types of headaches (7), and the possibility of new coexisting tension-type headaches (TTH) or cervicogenic headaches in patients doing remote work at home could contribute to a worsening during the lockdown cannot be ruled out.

A specific question about already having COVID-19 disease was done, and we had no infected patients in the survey, so the possibility of additional headaches due to COVID-19 disease has been ruled out in this sample.

Further, working from home impacted sleep patterns, which we know are vital in maintaining migraine control. We found that poor sleep has an impact on worsening migraine. Those who adapted their sleeping time and rested more hours improved their migraine (OR 5.6), unlike those who slept fewer hours or perceived poor sleep quality, and their migraine worsened (OR 6.1). In contrast, no association between the amount of coffee intake and the impact on migraine was found, which is classically described as a migraine trigger.

### 4.2. Access to Medical Services and Migraine

Those that needed it but had no access had three times more impact in worsening their symptoms than those that did not require it. (*p* < 0.0001). Patients with access worsened their symptoms 2.9 times more than those not needing it. Reflecting the described missed regular care and medications during the COVID-19 pandemic in the World Health Organization (WHO) survey [16], telemedicine development to replace in-person consults was lower, reaching no more than 60% in low-middle income countries, as there were no guidelines on how to provide medical care continuity in non-communicable diseases (NCD).

### 4.3. Medication Overuse in Patients Worsened Migraine Symptoms

Patients that used more acute medication for pain worsened 18 times more compared to those not increasing use or not needing more medication. Probably overuse of medication could lead these patients to Medication Overuse headaches. We found the opposite in those patients who did not increase the use of more drugs; that group had 38 times more chance of improving their migraine than those requiring the same amount of medicine.

### 4.4. Fear to COVID-19 Impact on Migraine

Due to the COVID-19 mortality, impact on health services, economic consequences, and government measures through social distancing and isolation, a significant psychological effect was caused across countries. This emotional response or fear of COVID-19 may impact migraine symptoms. To assess this phenomenon, we used the questionnaire about fear of COVID-19, which was answered similarly in all three countries. When comparing participants that worsened or improved with those that remained unchanged, significant differences in the total score were observed only for those whose migraine symptoms worsened. Thus, questions were included related to news of COVID-19 and social networks, the symptoms caused by COVID-19, and the uncertainty of when the pandemic could stop. These were found to be three factors that contributed to the worsening of migraine in the patients of the three countries [17].

The fear of COVID-19 during the initial months of the confinement, when the contagions had not yet become massive and the number of deaths increased, might have provoked increased uncertainty. Reports from the three countries for the third week of June 2020 showed that the estimated number of deaths in Mexico was around seventeen thousand, in Peru, about eight thousand, and in Argentina, about one thousand. However, by the fifth month of lockdown, there was an effect of fear of COVID-19 regarding worsening migraine symptoms as the pandemic advanced [12,17,18,19,20].

### 4.5. Limitations

Some limitations should be considered in our study. As the analysis was performed during the first wave of COVID-19 in Latin America, there were no guidelines or projection models to measure the impact of the COVID-19 pandemic on NCD or digital tools to record patient management to compare data before and during the pandemic period, including adapted validated tools for remote control of migraine or other types of headaches [21]. The cross-sectional design, with a subjective perception by the patients about worsening or improvement on migraine days, cannot be compared with previous and present data since neither used a migraine diary to assess the reliability of provided answers. Hence, a few publications using a migraine diary were performed in Latin American countries, including Chile and Argentina [22].

On the other hand, migraine studies with these characteristics referring to the COVID-19 Pandemic lockdown in Latin America and migraines have not been reported at the time of survey application. Additionally, the number of participants by each country, being that Argentina’s participants were a reduced group. Few previous studies on migraines as a disease are available regarding comparisons between Latin American Countries. In 2005, The Latin American Migraine Study completed questionnaires to determine the prevalence of migraine in urban communities [23]. In 2013, the Consensus on guidelines for chronic migraine treatment included some Latin American countries [19], and in 2016, the Project reported Medication Overuse Headaches in some Latin American countries [12].

In this study, the patients were evaluated for multiple possible causes regarding changes in migraine status instead of an evaluation of a single reason, which was subjected to several biases, mainly about the subjective character of the information provided by the participants.

Patients receiving the survey were already known migraine patients included in the study by neurologists with additional training in headaches. Most patients came from their headache clinics. There was a lack of medical advice for calendarizing their migraine days/month for this specific survey. Answers about the type of migraine, frequency of migraine in terms of #days/month, and treatment on preventive medication were not analyzed as a group. Different schemes for migraine prevention were used, according to each neurologist’s practice, all based on approved preventive medication drugs. This could be a limitation as results are not representative of migraine in the general population, although we believe it could be extrapolated to them.

However, we could mention the strength of our study. The total recruited questionnaires did not display essential differences across countries, which allows us to analyze sociodemographics, lifestyle, mood, and fear of COVID-19 and its effect on improvement or worsening migraine.

## 5. Conclusions

Access to health services was not a limitation for worsening or improvement in migraine patients, but medication intake was directly proportional to the worsening/improvement of migraine in all evaluated countries. We must keep working with our patients “that more is not better,” as this education on medical visit settings is mandatory to prevent medication overuse headaches.

Confinement during the first pandemic wave in patients of the three evaluated countries in Latin America worsened their migraine, especially for those staying at home during the lockdown, and adding exposure to continuous negative information in the media regarding COVID. Furthermore, we found a relationship between improving migraine symptoms, sleeping more hours than usual, and exercising.

Physicians should be aware of these findings to avoid future disruptions to migraine patients’ well-being in their daily practice and they also need to consider them in other similar situations and promote the use of migraine e-diary as a tool for remote control.

Confinement impacted Migraine Symptoms in migrainesSymptoms worsened 3.6 times for those who worked from homeIncreased intake of analgesics was associated with worsening of migraines up to 18 times more than those who remained with an unchanged intakeReduction of intake of analgesics has an OR 38.5 for improvement of symptoms compared to the unchanged group.News about COVID-19 and social media-related publications, the symptoms caused by COVID-19, and the uncertainty about when the pandemic will stop, were the three items of the Fear of COVID Questionnaire that contributed to the worsening of migraine in patients of the participating countries.

## Figures and Tables

**Table 1 ijerph-20-03598-t001:** Demographic of the respondents across Argentina, Mexico, and Peru.

Variables	Indicator	Total	Mexico	Peru	Argentina
		(n = 243)	(n = 99)	(n = 116)	(n = 28)
Age (years)					
	<30 years-old	50 (20.6%)	16 (16.2%)	27 (23.3%)	7 (0.25%)
	30–50 years-old	127 (52.3%)	50 (50.5%)	61 (52.6%)	16 (0.571%)
	>50 years-old	66 (27.2%)	33 (33.3%)	28 (24.1%)	5 (0.179%)
Gender					
	Female (n, %)	207 (85.2%)	80 (80.8%)	103 (88.8%)	24 (0.857%)
Civil status					
	Cohabiting (n, %)	145 (59.7%)	69 (69.7%)	60 (51.7%)	16 (0.571%)
Occupation before lockdown					
	Out of home (n, %)	198 (81.5%)	84 (84.8%)	87 (75%)	27 (0.964%)
Months in lockdown					
	2–8 (n, %)	189 (77.8%)	66 (66.7%)	103 (88.8%)	20 (0.714%)
Still in lockdown					
	Yes (n, %)	208 (85.6%)	77 (77.8%)	109 (94%)	22 (0.786%)
Lockdown restriction					
	Yes (n, %)	120 (49.4%)	38 (38.4%)	73 (62.9%)	9 (0.321%)

**Table 2 ijerph-20-03598-t002:** Multinomial logistic regressions for improvement and worsening migraine (symptoms remained unchanged as reference) and demographic characteristics of patients, type of confinement, changes caused by confinement, emotional status, and fear of COVID-19.

			Improvement				Worsening		
Variable	Indicator	B	OR	95%IC	*p*-Value	B	OR	95% CI	*p*-Value
**Demographic characteristics**									
Civil status	Single (reference)								
	In a relationship	0.181	1.199	(0.483, 2.974)	0.696	−0.245	0.783	(0.388, 1.579)	0.493
Age	50+ age (reference)								
	30− age	−0.144	0.866	(0.477, 1.574)	0.407	−0.403	0.668	(0.406, 1.00)	0.5
	30–50 year-old	0.781	2.184	(0.871, 5.477)	0.096	0.717	2.049	(1.05, 3.999)	0.035 *
**Type of confinement**									
Months in lockdown	Less than two months (reference)								
	2–8 months	−0.511	0.6	(0.188, 1.91)	0.387	−0.269	0.764	(0.308, 1.895)	0.561
Still in lockdown	No (reference)								
	Yes	−0.222	0.801	(0.218, 2.945)	0.739	−0.088	0.916	(0.322, 2.603)	0.869
Lockdown restriction	No (reference)								
	Yes	0.101	1.106	(0.481, 2.545)	0.812	0.134	1.144	(0.621, 2.107)	0.666
Work setting	Out of home (reference)								
	At home	−1.369	0.254	(0.031, 2.094)	0.203	1.321	3.746	(1.68, 8.354)	0.001 *
**Changes caused by confinement**									
Sleep	Remained unchanged (reference)								
	Less quality	−0.142	0.867	(0.502, 1.499)	0.61	0.861	2.366	(1.386, 4.038)	0.002 *
Exercise	No (reference)								
	Increased Exercise	0.957	2.604	(1.498, 4.525)	0.001 *	0.253	1.288	(0.783, 2.116)	0.319
Daily routine	Unchanged (reference)								
	Adjusted to lockdown	0.919	2.507	(1.087, 5.786)	0.031 *	−0.04	0.961	(0.474, 1.949)	0.912
Analgesic abuse	No (reference)								
	Increased consumption	−0.588	0.556	(0.161, 1.921)	0.353	2.701	14.889	(7.075, 31.334)	0.000 *
**Emotional status**									
Anxiety/depression	Absent (reference)								
	Increased symptoms	−0.147	0.863	(0.692, 1.077)	0.192	0.204	1.226	(1.062, 1.416)	0.006 *
**Fear to COVID-19**		0.041	1.042	(0.969, 1.12)	0.266	0.056	1.058	(1.002, 1.116)	0.042 *

B, coefficient “B”. OR, odds ratio, “Exp(B)”. * χ^2^ test, statistically significant, *p* < 0.05.

## Data Availability

The data sets used and analyzed during the current study are available from the corresponding author upon reasonable request.

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
