# Peer review of "Impact of COVID-19 Pandemic Lockdown on Migraine Patients in Latin America"

_ijerph, 2023, doi:10.3390/ijerph20043598_

Round 1

Reviewer 1 Report

Congratulations and thanks to the authors for the effort made during the pandemic over this relevant issue. My greatest question is if all the patients were evaluated by a trained neurologist prior to the application of the questionnaire (1 or 2 weeks before), in order to corroborate any other type of headache coexisting with migraine, like tension-type headache (TTH), that many times the patient may qualify as migraine. A phenomenon that occurred during and after the pandemic, due to stress factors, depression, remote work, etc. the incidence of Tension-type headache (TTH) or cervicogenic headache increased, these types of headache can coexist with migraine symptoms, and the patients may consider these pains as migraine easily; and only a specialized and timely evaluation could differentiate it and avoid classifying any pain as migraine.

Author Response

Dear Reviewer, 

We add a specific paragraph answering your question: 

"All trained neurologists with medical practice on headache patients as well as general neurology patients, and with full knowledge of the current International Classification of Headache Disorders (ICHD-3) [7]. Those who wanted to participate needed to send an online questionnaire-based survey to their current migraine-diagnosed patients."

We also improved the writing of the paper. Please see attached.

Reviewer 2 Report

Thank you for letting me evaluate this paper from a study of the consequences of the covid-19 pandemic and related measures on migraine patients across three Latin-American countries. The results are important not just for understanding the effects of the pandemic itself but also for furthering the understanding of the importance of health service support vs well as lack of it and factors associated with isolation and psychosocial setting for migraine patients.

The stated objective of the study appears to be to evaluate pandemic related trigger factors for worsening of migraine. In addition, improved behavioural responses during the pandemic and differences between the three countries involved are focused.

 I enjoyed reading the paper and find the results important and deserving of a wider spread. I hope I can help add some suggestions that may help the authors improve the paper even further.

Major points: 

1. Objective: I suggest this could be stated more stringently and perhaps be more clearly formulated. The first part of the sentence is clear enough but it is not easy to understand what is meant by "improving behavioural responses between the L Am participating countries" and, indeed, how this objective could be measured. 

2. Methods:

i) I lack at least some basic information about the migraine characteristics of the participants - at the very least in order to interpret the chronic migraine/MOH spectrum some information about migraine frequency (no days/month) is needed. Also no information is given on preventive medication. 

ii) Statistical analyses: Please add more information on the statistical models used - it seems that a plural of "models" were used - specify the analytical models in terms of included variables.

3. Results: The tables in their present form are far from optimal and the poor formatting especially of table 1 makes it basically unreadable - this is a basic requirement which may be partly due to journal submission policies but in this case complete separate tables in the original format are far preferable to tables embedded in the document.

The results of the multivariable logistic regression models are unclearly reported - are the analyses presented from one single model with all variables included? Or are perhaps several different models analysed as one is led to believe in the results reporting section in the text on page 7 where covariates are presented in different groups. If the latter, then all models should be presented either in separate tables or in a table synthesis but with the variable included in each model clearly delineated.

4. Discussion: I suggest the discussion needs a thorough work through. It needs to be shortened in general and reiteration of results should be dropped except perhaps for summarising the main findings to be discussed. Finally, speculative discussion without direct evidence from the study or other references should be dropped. For example lines 279 to 282 should be removedas they are neither based on own evidence or on referred evidence of others. Similarly lines 286 to 290 consist of unsubstantiated and un-referred value judgements that do not add to the interpretation of the main results. 

The four numbered results points discussed on page 9-11 are not based in the stated objectives. I suggest you can be more stringent and congruent here. 

I suggest to emphasise the main results ie the factors identified during the pandemic which predicted worsening (or improvement) of migraine. Your findings of the very strong association and large effect size of increased (and reduced) acute medication are highly relevant (18 x OR for worsening!), especially coupled with factors such as reduced health service contacts and isolation.  Such findings are very important for migraine care and need consideration in other similar situations.

Limitations: Add the limitation of patients recruited via headache clinics not being representative of migraine in the general population

Minor:

English language proof reading could in some cases improve understanding.

Author Response

Dear Reviewer,

Thank you for your comments. We uploaded a revised manuscript with included paragraphs in red. Please check the attached document.

Sincerely

Reviewer 3 Report

I would improve this beautiful study by adding the following references (and maybe one accompanying sentence for each in the text):

- Gelfand AA, Poland G. Migraine treatment and COVID-19 vaccines: No cause for concern. Headache. 2021 Mar;61(3):409-411. doi: 10.1111/head.14086. PMID: 33543775; PMCID: PMC8013648.

Angus-Leppan H, Guiloff AE, Benson K, Guiloff RJ. Navigating migraine care through the COVID-19 pandemic: an update. J Neurol. 2021 Nov;268(11):4388-4395. doi: 10.1007/s00415-021-10610-w. Epub 2021 May 17. PMID: 34002281; PMCID: PMC8128091.

- (for a general sentence on the importance of prevention of the covid-19 pandemic): Galassi FM, Pate FD, You W, Gurr A, Lucas T, Antunes-Ferreira N, Varotto E, Habicht ME. Pandemic realism as the indispensable political precondition for global disease eradication. Public Health. 2022 Nov;212:55-57. doi: 10.1016/j.puhe.2022.08.016. Epub 2022 Oct 7. PMID: 36215929.

Author Response

Dear Reviewer,

We modified the manuscript as you suggest. Please check the attached document.

Sincerely, 
